# MST-GNN: Graph Neural Network with Multi-Granularity in Space and Time for Traffic Prediction

## Abstract

Traffic flow prediction based on vehicle trajectories is a crucial aspect of Intelligent Transportation Systems (ITS). Deep learning approaches have recently been widely adopted to capture spatio-temporal correlations in traffic conditions, and have achieved superior performance compared to traditional methods. However, most existing studies focus on traffic prediction at a single spatial scale, usually corresponding to the road-segment level. According to the Hierarchy Theory, processes at different scales form a hierarchy of organization, and meaningful patterns may emerge at multiple levels of details. Presetting traffic data at an inappropriate scale can cause misunderstanding in features learning. In this paper, we propose a graph learning model called MST-GNN, which captures the comprehensive behaviors and dynamics of traffic patterns by fusing multi-scale features from both space and time perspectives. In ITS applications, users usually consider traffic conditions at the larger-scale regional level, and a prediction model must attend to multi-scale application requirements. Moreover, the structure of multiple granularities in time series can fully unleash the potential of different temporal scales in learning dynamic traffic pattern features. We inject the multi-scale spatio-temporal structure into a graph neural architecture with a tailored fusion module. Our model achieves state-of-the-art accuracy prediction results on two traffic benchmark datasets.

## 1 Introduction

Traffic prediction is a special time-series forecasting, whose main task is to predict the future traffic conditions based on historical observations. It is a critical ingredient of the intelligent transportation system (ITS), which has great effect on the fields of efficient management of urban traffic, rational allocation of traffic resources, and planning of urban construction Zhang et al. (2011).

Traffic flow prediction is not a new task, early attempts at prediction were made using statistical methods EDES et al. (1980); Ahmed & Cook (1979). More recently, some scholars have used machine learning methods such as support vector regression (SVR), K nearest neighbor (KNN) to build prediction models Castro-Neto et al. (2009); Hu et al. (2016); Davis & Nihan (1991); Zheng et al. (2020b). However, due to the complex spatio-temporal characteristics of traffic flow data, statistical and machine learning models have limitations and cannot be adapted to more complex application scenarios. The development of deep learning technology has brought a new turn for the problem of traffic flow forecasting. A large number of research results have proved that the deep learning method is currently the optimal solution to the problem of traffic flow forecasting Wu et al. (2020); Fan et al. (2020); Jiang & Luo (2022). Recurrent neural networks (RNN, GRU, LSTM, etc.) were first verified effective in traditional time series prediction, and are often used in traffic prediction models to capture temporal features Wu & Tan (2016); Zhao et al. (2019). Meanwhile, in order to capture the adjacent relationship between traffic nodes, researchers successively introduced a convolutional neural network (CNN) Wu & Tan (2016); Henaff et al. (2015) suitable for Euclidean structure data and a graph neural network (GNN) Agafonov (2020); Guo et al. (2020); Chen et al. (2020) suitable for graph structure data. Recently, some models have used spatio-temporal attention mechanism, which also achieved excellent accuracy Su et al. (2022); Shi et al. (2020); Do et al. (2019); Guo et al. (2019); Zheng et al. (2020a).

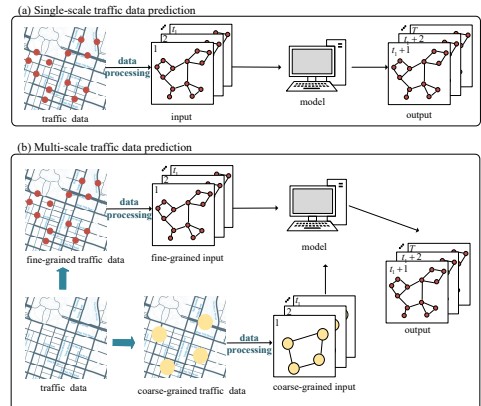

Figure 1: Schematic diagram of multi-spatial-scale data. (a) Single-scale traffic data prediction example. (b) Multi-scale traffic data prediction example.

The above studies mainly focus on the traffic flow changes at a single spatial scale He & Shin (2020); Wang et al. (2022); Liao et al. (2022), or only mine the correlations at a single temporal scale Wang et al. (2020); Guo et al. (2021); Mallick et al. (2020). Although the traffic flow forecasting research unceasingly obtains the new breakthrough, some deficiencies remain unresolved. The existing models still face challenges in the traffic features learning over long distances and long periods. We argue that there are still two important aspects ignored in these approaches.

**Ignoring the integration of traffic features across spatial scales** Many recent forecasting models have focused on traffic forecasting at the road segment scale (which can be considered as fine-grained), while ignoring the traffic conditions at the urban regional scale (i.e., is coarse-grained). However, in ITS applications, traffic forecasting at the urban regional scale is also indispensable, which can help better observe the macroscopic traffic conditions, and allocate urban traffic resources effectivelySun et al. (2016); Ding et al. (2019). As shown in Figure 1, there is a certain correspondence between coarse-grained data and fine-grained data, and the fine-grained data often generates smaller features regarding spatial correlation. Combined with the characteristics of coarse-grained data, it can assist fine-grained process to learn more macro traffic information (e.g., longer-distance dependency), thereby improving the prediction accuracy.

**Ignoring the integration of traffic features across time scales** As shown in Figure 2, the characteristic information contained in flow data of different time scales is not same but compensates each other. Existing forecasting models tend to ignore such multi-scale information interaction. In training, fine-grained data often requires multi-step iterations for learning long-term dependency features (e.g., A−>B as shown in Figure 2a). In contrast, under the coarse-grained series, the number of iterations required to propagate the information of the same cycle can be greatly reduced (e.g., A−>B as shown in Figure 2b). While reducing the number of propagation layers, the model can effectively maintain the critical information during the iterative process.

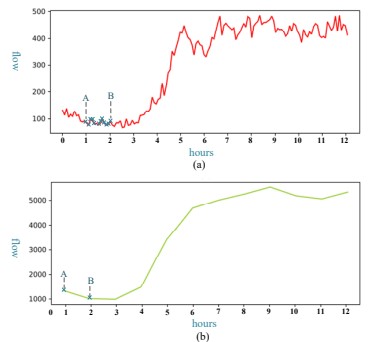

Figure 2: Comparison of flow changes on different time scales. The sampling time step of panel (a) is 5 minutes, and the sampling time step of panel (b) is 1 hour.

Overall, meaningful patterns may emergency at multiple levels of details, and the Hierarchical Theory O'Neill et al. (1989) states that smaller-scale processes sustain larger-scale processes, while the larger processes constrain the smaller-scale processes by setting the boundary conditions. Therefore, cross-scale integration of traffic information can be a potential contributor to advance the traffic flow forecasting research.

For the above content, we propose a Multi-granularity Spatio-Temporal Graph Neural Network (MST-GNN). In this model, we input the traffic data of multiple spatial scales, among which the coarse-grained data is obtained through the community detection. Besides, we add time series information on a long-term scale to supplement the model with the dynamic characteristics of the traffic flows at the macro level. The main contributions of this paper are as follows:

- An effective framework is proposed to simultaneously capture multi-temporal-scale and multi-spatial-scale information by designing a multi-scale spatial-temporal data mining module.

- Based on the spatio-temporal correlation of multi-temporal-scale data, a spatio-temporal attention module that fuses multi-temporal-scale information is proposed.

- Despite MST-GNN's backbone being a simple model, our method still surpasses existing methods and achieves SOTA performance on benchmark datasets.

To the best of our knowledge, this is the first work to demonstrate the exceptional ability of multi-scale spatio-temporal integration to traffic prediction with remarkable accuracy. We expect this can serve as an effective tool for more various tasks in the spatial-temporal data mining field, in which scale is an inevitable part of features learning.

## 2 RELATED WORK

### 2.1 TRAFFIC PREDICTION

Traffic flow data is a typical spatio-temporal data that exhibits complex spatio-temporal features. Recently, traffic forecasting researches rely on deep learning models to model spatio-temporal features due to its superior performance. Yu et al. (2017) proposed the STGCN model, which combines the graph convolutional layer with the convolutional sequence learning layer to model the spatiotemporal dependencies of the road network. Zhao et al. (2019) combined the GCN with the gated recurrent unit (GRU) to build the T-GCN model. What's more, some researchers have tried to introduce the mature methods that have been applied in other time series forecasting topics into the traffic forecasting model and have made new breakthroughs. Li & Zhu (2021) proposed the SFTGNN model, which introduced a dynamic time warping (DTW) method to generate a time graph to compensate for the shortcomings of the spatial graph in the traditional model. Based on neural control differential equations (NCDEs), Choi et al. (2022) build the STG-NCDE model that can implement the spatial and temporal features mining effectively. However, the above methods only focus on the single-scale spatio-temporal features of traffic data.

### 2.2 SPATIO-TEMPORAL ATTENTION MECHANISM

Compared with traditional time-series forecasting, traffic flow forecasting is more complicated, especially in long-term or long-range tasks. The attention mechanism originated in the field of computer vision, and has been applied to many deep learning frameworks due to its flexibility and practicality Vaswani et al. (2017); Guo et al. (2022); Niu et al. (2021). In order to release the challenge of learning long-term dependence of traffic data, some researchers introduce attention mechanism into predictive models Su et al. (2022); Shi et al. (2020); Do et al. (2019). Guo et al.Guo et al. (2019) proposed an attention-based spatio-temporal graph convolutional network ASTGCN, which contains a temporal attention layer and a spatial attention layer to capture the spatio-temporal correlation between traffic data. Zheng et al. Zheng et al. (2020a) proposed a graph multi-attention network GMAN, which adopts the architecture of encoding and decoding. An attention conversion module is included between the encoder and decoder to simulate the relationship between historical time steps and future time steps, which helps to alleviate the problem of error propagation between prediction time steps. Despite the usefulness of attention mechanism in these researches, effectively learning the long-range dependency from both perspectives of space and time remains a challenging task.

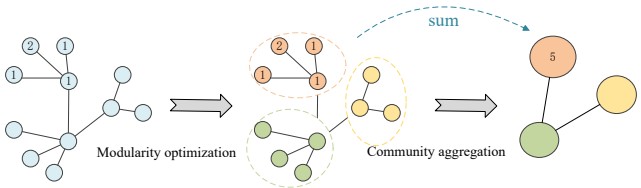

Figure 3: Schematic diagram of the Louvain algorithm

## 2.3 COMMUNITY DETECTION

Due to the efficiency and accuracy of Louvain algorithm, it has become the most widely used method in community detectionClauset et al. (2004); Wakita & Tsurumi (2007); Pons & Latapy (2005). The Louvain algorithmBlondel et al. (2008) is a designed modularity calculation for community detection. It is mainly divided into two stages (Figure 3 ): modularity optimization and network cohesion. Modularity is a measure of the quality of community detection. The higher the community modularity, the tighter the internal nodes of the same community in the divided graph structure are, and the higher the community detection quality is. The formula for calculating the modularity is as follows:

$$Q = \frac{1}{2m} \sum_c [\sum in - \frac{(\sum tot)^2}{2m}] \tag{1}$$

where $\sum in$ indicates the sum of weights of edges in community $c$, $\sum tot$ indicates the sum of weights of edges connecting all nodes in community $c$, $m$ ndicates the sum of weights of all edges in the whole graph. The Louvain algorithm is used in the detection of various networks. Li et al. (2022) proposed a new traffic area division method based on the Louvain algorithm, which can dynamically divide the regional road network according to the changes of traffic features. Compared with traditional methods, the division results of this model are more objective. Zhang et al. (2021) used the Louvain algorithm to divide the urban community structure, laying the foundation for mining the urban core traffic area. In this paper, the Louvain algorithm is used to obtain traffic data at the urban region scale.

## 3 METHODOLOGY

As shown in Figure 4, our model involves multi-spatial-scale fusion module, multi-temporal-scale fusion module and output fusion layer. In the following, we will elaborate on the above contents in four parts.

### 3.1 DATA PREPROCESSING

The data preprocessing required by the model includes two parts: data generation at a regional scale and data generation at a longer time scale.

The data generation at the regional scale is mainly realized by the Louvain community detection algorithm, and the implementation process is shown in Figure 3. Suppose the input original fine-grained spatial data is $G_f$, the coarse-grained spatial data obtained by the Louvain algorithm is $G_c$, and the mapping matrix between them is $A_{fc}$. For node $G_c$ in $n_i$, its eigenvalue is the sum of the eigenvalues of all the fine-grained nodes it contains.

Figure 2 shows the traffic data changes of the same node on different time scales within twelve hours. Let the original data be fine-grained data, denoted as $X_{T_f} = \{x_{T_f}^1, .., x_{T_f}^t, ..., x_{T_f}^{T_1}\}$, where $T_1$ is the input time step. Coarse-grained data is summed from fine-grained data, denoted as $X_{T_c} = \{x_{T_c}^1, .., x_{T_c}^t, ..., x_{T_c}^{T_1'}\}$, where $T_1'$ is the time step of the input. For coarse-grained data $x_{T_c}^t$,

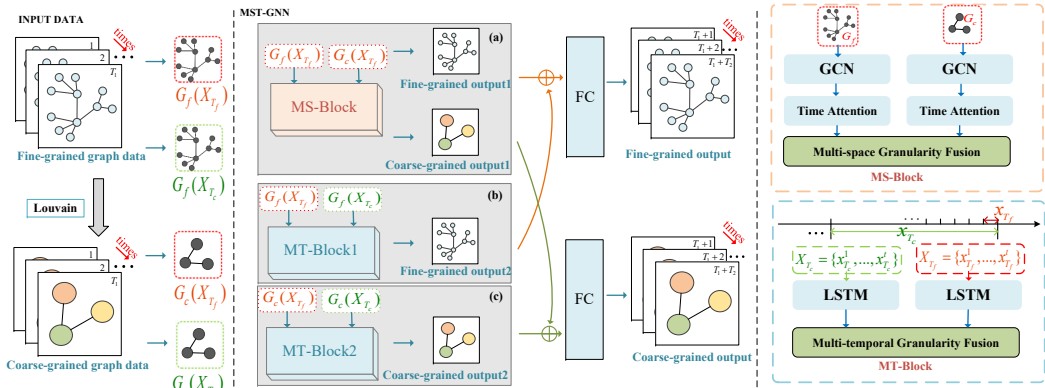

Figure 4: The framework of Multi-granularity Spatio-Temporal Graph Neural Network (MST-GNN). (a)The multi-spatial-scale neural network (MS-Block) mainly deals with the fusion of multi-spatial-scale data, and the input is multi-spatial-scale data at a smaller temporal scale. The multi-temporal-scale neural network (MT-Block) mainly deals with the fusion of multi-temporal-scale data, and the input is fine-grained spatial units (b) and multi-time-scale data (c) under coarse-grained spatial units.

its evaluation formula is shown as follows:

$$x_{T_c}^t = \sum_{i=t-n*p}^{t} x_{T_f}^i \tag{2}$$

$$p = T_c/T_f \tag{3}$$

Among them, $n$ is the time step of a single input coarse-grained data. $T_c$ is the temporal scale of the input coarse-grained data, which is 1 hour as shown in Figure 2. $T_f$ is the temporal scale of the input fine-grained data, which is 5 minutes. $T_c$ must be an integer multiple of $T_f$, that is, $p$ must be an integer. Furthermore, for Equation 2, we stipulate that it must satisfy $(t - n * p) > 0$.

## 3.2 MULTI-SPATIAL-SCALE FUSION MODULE

The multi-spatial-scale fusion module (MS-Block) is designed to process and fuse the spatiotemporal features of multi-spatial-scale data, mainly including a graph neural network (GCN) layer, a temporal attention layer and a multi-scale features fusion layer. This module is inputted with data of different spatial granularities under fine time scale. First, it extracts spatial features from the two scales through the GCN layer, then extracts the temporal features of the data through the self-attention layer, and finally aggregates the data features at multiple spatial scales through the fusion module.

**Graph Neural Network Layer:** In the model, we use a graph convolutional neural network (GCN) to learning the spatial features of traffic dataKipf & Welling (2016). For graph structure $G = (V, E, A)$, let $H^{(l)}$ be the current eigenvector matrix of all nodes, then the formula of a convolution operation can be expressed as follows:

$$H^{(l+1)} = \sigma(\widetilde{D}^{-\frac{1}{2}} \widetilde{A} \widetilde{D}^{-\frac{1}{2}} H^{(l)} W^{(l)}) \tag{4}$$

where $\widetilde{A} = A + I_N$, $I_N$ is the identity matrix. $\widetilde{D}$ is a degree matrix, and the calculation method is $\widetilde{D} = \sum \widetilde{A}_{ij}$. $\sigma(\cdot)$ is the activation function.

**Temporal Attention Layer:** The temporal attention layer in the model is realized by performing self-attention processing on the input multi-scale data in the time dimension, and its main function is to learn the temporal embedding of the data. After passing through the GCN layer, the current input data is $X^l$. First, the query $(Q^l)$, key $(K^l)$ and value $(V^l)$ are obtained through linear transformation,

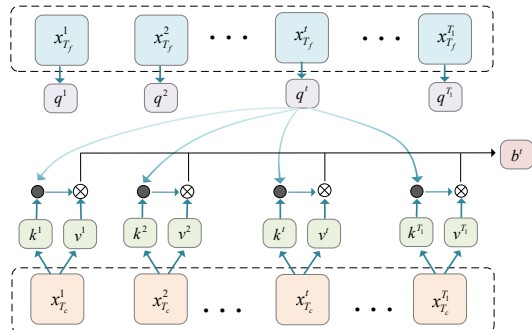

Figure 5: Multi-temporal-scale data fusion module based on spatial attention.

and then the weight of the $V$ elements corresponding to the key value is obtained by calculating the similarity of elements in $Q$ and $K$, and the final output result is obtained by weighted summation.

$$Q^l = X^l \cdot w_q \tag{5}$$

$$K^l = X^l \cdot w_k \tag{6}$$

$$V^l = X^l \cdot w_v \tag{7}$$

$$O^l = softmax(Q^l \cdot (K^l)^T \cdot V^l) \tag{8}$$

where $O$ is the output of this calculation. $w_q$, $w_k$ and $w_v$ are trainable parameters.

**Multi-scale Features Fusion Layer:** MS-Block takes both coarse-grained spatial data $G_c$ and fine-grained spatial data $G_f$ as the input. After the GCN layer and the temporal attention layer, we use a multi-scale features fusion layer to fuse the data features of the two scales. In this layer, the spatial features of the multi-scale data interact with each other through a mapping matrix of the two sets of data.

$$X_c^{l+1} = X_c^l + \eta_1 \cdot softmax(A_{fc} X_f^l W_c) \tag{9}$$

$$X_f^{l+1} = X_f^l + \eta_2 \cdot softmax(A_{fc}^T X_c^l W_f) \tag{10}$$

where $X_c^l$ and $X_f^l$ are the input coarse-grained data and fine-grained data respectively. $\eta_1$ and $\eta_2$ are hyperparameters. $A_{fc}$ is the mapping matrix obtained in the data preprocessing stage. $W_c$ and $W_f$ are trainable parameters.

### 3.3 MULTI-TEMPORAL-SCALE FUSION MODULE

The multi-temporal-scale module (MT-Block) aims to process and fuse the spatio-temporal features of traffic data at multiple time scales, including an LSTM layer and a multi-scale fusion layer based on spatial attention. In this module, the input is the traffic data of multiple time scales under the same spatial scale, and the output is finer-temporal-scale data.

**LSTM Layer:** LSTM can solve the long-term dependence learning problem of cyclic neural networkMa et al. (2015). It has good performance in the processing of time series data. In this paper, LSTM is used to extract temporal features of traffic data at different time scales. For the input time series $X^l \in \mathbb{R}^{s \times t \times f}$, the LSTM model extracts the embedding $O \in \mathbb{R}^{s \times t \times f}$. Then it goes through a fully connected layer for linear transformation to get the output $X^{l+1} \in \mathbb{R}^{s \times n \times f}$ of the LSTM layer. Among them, $s$ represents the spatial dimension of data, $t$ represents the time step of data input, $n$ represents the time step of data output, and $f$ represents the feature dimension of traffic data.

**Multi-scale Fusion Layer:** After the input traffic data of multiple time scales are processed by the LSTM layer, two sets of coarse-grained and fine-grained outputs are obtained. The multi-temporal-scale fusion module is designed on the spatial attention, and the specific structure is shown in Figure 5. By calculating the similarity between coarse-grained data and fine-grained data, the weight value is determined, and then the coarse-grained data is weighted and summed to obtain the fusion value

of coarse-grained data relative to the fine-grained data. The calculated fusion value is added to the output of the fine-grained data of the previous layer to obtain the final output of the fusion module. The calculations are as follows:

$$Q^l = X_f^l \cdot w_q' \tag{11}$$

$$K^l = X_c^l \cdot w_k' \tag{12}$$

$$V^l = X_c^l \cdot w_v' \tag{13}$$

$$O^l = softmax(Q^l \cdot (K^l)^T \cdot V^l) \tag{14}$$

$$X_f^{l+1} = O^l + X_f^l \tag{15}$$

where $X_f^l$ and $X_c^l$ are the fine-grained and coarse-grained inputs of this layer respectively. $X_f^{l+1}$ is the output of fine-grained data in this layer. $w_q'$, $w_k'$ and $w_v'$ are all trainable parameters.

### 3.4 Output Fusion Layer

As shown in Figure 4, the MST-GNN contains one MS-Block module and two MT-Block modules. MS-Block simultaneously outputs coarse-grained spatial data $X_c^s$ and fine-grained spatial data $X_f^s$. The two MT-Block modules output coarse-grained spatial data $X_c^t$ and fine-grained spatial data $X_f^t$ corresponding to the input data scale, and all the output data have fine-grained time scales.

The output fusion layer fuses the output results under the same spatial scale respectively, and integrates the multi-temporal features and spatial features to obtain the final output result. Specifically: First, the output results under the same spatial scale are spliced in the time dimension to obtain a longer vector. Then, after a fully connected layer, the longer vector is linearly transformed to obtain the final output result. The calculation formulas are as follows:

$$X_{f\_out} = FC([X_f^s, X_f^t]) \tag{16}$$

$$X_{c\_out} = FC([X_c^s, X_c^t]) \tag{17}$$

### 3.5 Loss Function

Note that the fine- and coarse-grained predictions should be consistent. We use the loss function for the two prediction tasks as follows:

$$\mathcal{L} = \eta_1 ||X_{f\_target} - X_{f\_out}||^2 + \eta_2 ||X_{c\_target} - X_{c\_out}||^2 \tag{18}$$

$X_{f\_target}$ and $X_{c\_target}$ are the target values of the corresponding spatial scale prediction tasks, respectively. $\eta_1$ and $\eta_2$ are hyperparameters, and we set them both to 0.5 in the experiment.

## 4 Experiments

### 4.1 Dataset

Our experiments are carried out on two benchmark datasets (California highway datasets), PeMSD4 and PeMSD08Jia et al. (2001). The data is aggregated every five minutes, that is, each sensor contains 288 pieces of data per day. The traffic variables include flow only. The PeMSD4 dataset contains 307 sensors, and the time span is from January 2018 to February 2018. The PeMSD8 dataset contains 170 sensors and the time span is from July 2016 to August 2016. In the experiments, we use the first 60% of the data as the training set, 20% of the data as the validation set, and the last 20% of the data as the test set.

After being processed by the Louvain community discovery algorithm, the PeMSD4 and PeMSD8 data sets contain 31 communities and 15 communities respectively. In addition, we preprocess the data sets by normalizing the maximum and minimum values, and then the data values are between 0 and 1. The specific formula is that $x' = (x - x_{min})/(x_{max} - x_{min})$, where $x_{min}$ and $x_{max}$ are the maximum and minimum values in the corresponding test set, respectively.

Table 1: Performance comparison of MST-GNN and baseline models on PeMSD4 dataset.

| Time | 5mins | | | 15mins | | | 30mins | | |
|------|-------|------|------|--------|------|------|--------|------|------|
| Metric | MAPE | MAE | RMSE | MAPE | MAE | RMSE | MAPE | MAE | RMSE |
| GCN | 0.213 | 28.18 | 39.68 | 0.219 | 29.29 | 41.13 | 0.227 | 30.60 | 42.89 |
| LSTM | 0.138 | 21.15 | 31.82 | 0.140 | 21.65 | 31.39 | 0.154 | 23.55 | 34.13 |
| TGCN | 0.192 | 23.59 | 35.01 | 0.194 | 23.79 | 35.37 | 0.221 | 25.83 | 37.90 |
| STGCN | 0.149 | 19.79 | 30.80 | 0.147 | 20.84 | 32.63 | 0.173 | 22.67 | 35.13 |
| ASTGCN | 0.201 | 30.97 | 51.13 | 0.181 | 26.47 | 42.17 | 0.184 | 27.17 | 43.29 |
| HGCN | 0.125 | 17.73 | 28.10 | 0.140 | 19.68 | 29.41 | 0.154 | 20.67 | 32.27 |
| STG-NCDE | 0.120 | 18.50 | 29.28 | 0.123 | 18.86 | 30.03 | 0.133 | 19.49 | 30.99 |
| **MST-GNN** | **0.098** | **14.51** | **23.22** | **0.107** | **15.69** | **25.06** | **0.120** | **17.44** | **27.73** |

Table 2: Performance comparison of MST-GNN and baseline models on PeMSD8 dataset.

| Time | 5mins | | | 15mins | | | 30mins | | |
|------|-------|------|------|--------|------|------|--------|------|------|
| Metric | MAPE | MAE | RMSE | MAPE | MAE | RMSE | MAPE | MAE | RMSE |
| GCN | 0.210 | 34.02 | 46.09 | 0.208 | 34.76 | 46.98 | 0.212 | 35.49 | 48.00 |
| LSTM | 0.099 | 18.00 | 26.61 | 0.121 | 21.69 | 29.96 | 0.116 | 20.96 | 29.88 |
| TGCN | 0.244 | 22.80 | 31.82 | 0.230 | 22.46 | 31.52 | 0.250 | 23.42 | 33.08 |
| STGCN | 0.125 | 16.24 | 24.93 | 0.134 | 17.45 | 26.70 | 0.143 | 19.53 | 29.60 |
| ASTGCN | 0.143 | 21.35 | 34.88 | 0.138 | 20.06 | 32.70 | 0.142 | 20.65 | 33.72 |
| HGCN | 0.092 | 13.32 | 20.44 | 0.102 | 14.66 | 22.62 | 0.109 | 15.75 | 24.66 |
| STG-NCDE | 0.096 | 14.27 | 21.92 | 0.090 | 13.73 | 21.55 | 0.107 | 15.98 | 24.81 |
| **MST-GNN** | **0.063** | **9.98** | **15.26** | **0.067** | **10.86** | **16.89** | **0.075** | **11.89** | **18.57** |

## 4.2 BASELINES

To test the performance of our proposed model, we selected multiple models for comparative experiments: (1) classic deep learning baseline models including GCNKipf & Welling (2016) and LSTMMa et al. (2015); (2) classic graph neural network models for traffic flow forecasting, including TGCNZhao et al. (2019) and STGCNYu et al. (2017); (3) recent graph neural network model for traffic flow prediction such as the attention mechanism based ASTGCNGuo et al. (2019); (4) recent graph neural network model with multi-scale information HGCNGuo et al. (2021); (5) short-term traffic flow forecasting model STG-NCDEChoi et al. (2022).

## 4.3 EXPERIMENTAL SETTING

In the experiment, one-hour historical data with 12 continuous time steps is used to predict next 1-time-step data, 3-time-steps data and 6-time-steps data. We have evaluated MST-GNN more than 10 times on each public dataset. Experiments are conducted under the environment with one Intel Core i9-10940X CPU @ 3.30GHz and NVIDIA GeForce RTX 3080 Ti card. We train our model using Adam optimizer with learning rate 0.0001. The performances of all methods are measured by three metrics, i.e., Mean Absolute Percentage Error (MAPE), Root Mean Square Error (RMSE) and Mean Absolute Error (MAE).

## 4.4 EXPERIMENT RESULTS AND ANALYSIS

Table 1 and table 2 shows the performance of the MST-GNN model and other baseline models on the PEMS04 and PEMS08 datasets. In the comparative experiment, we predict the traffic data of the next 5 minutes, 15 minutes, and 30 minutes respectively. For each notable model, we list its average MAE/RMSE/MAPE from the two datasets. All models experience some decrease in prediction accuracy as the time step of the forecast data increases. Compared with all models, our model has a 1.3% 11.5% accuracy improvement on PEMSD4 and a 2.3% 14.7% accuracy improvement on PEMSD8. The MST-GNN model performs best in all three indicators of MAPE/MAE/RMSE.

As shown in the table 1 and table 2, MST-GNN shows the best performance on all metrics. Experimental results demonstrate that the proposed multi-scale learning framework can effectively fuse spatio-temporal features across scales to improve prediction accuracy.

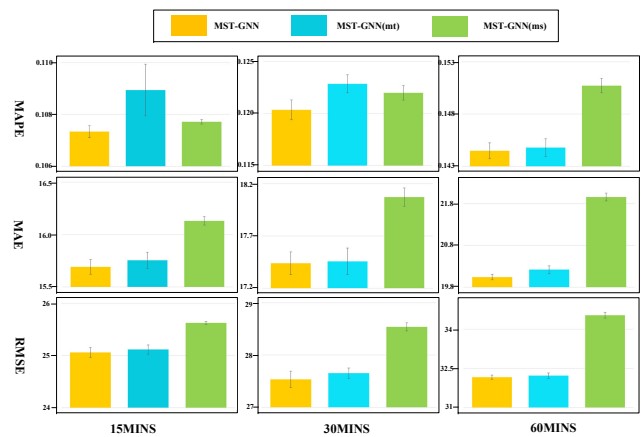

Figure 6: Ablation experiment comparison on PEMSD4 and PEMSD8.

## 4.5 COMPUTATION COST

In order to test the prediction efficiency of the model, we tested the training computation time and inference computation time of HGCN, STG-NCDE and MST-GNN, which have the highest comprehensive prediction accuracy in comparative experiments. The test data set is PEMSD4. In this experiment, one-hour historical data with 12 continuous time steps is used to predict next 1-time-step data. The results are shown in Table 3. Although MST-NCDE and HGCN also have excellent performance in prediction, the computation time of training is much higher than that of the MST-GNN model. In general, MST-GNN can obtain higher prediction accuracy in a shorter training time, that is, the prediction efficiency is the highest.

Table 3: The computation time on PeMSD4 dataset.

| Method | Training Computation Time (s/epoch) | Inference Computation Time (s) |
|---|---|---|
| HGCN | 37.3 | 4.12 |
| STG-NCDE | 426.5 | 42.7 |
| MST-GNN | 4.1 | 0.45 |

## 4.6 ABLATION STUDY

To verify whether the proposed multi-temporal-scale module and multi-spatial-scale module are effective, ablation experiments are carried out by comparing the complete model and the only-multiple-spatial-scale model called MST-GNN(ms) and only-multiple-temporal-scale model called MST-GNN(mt). We carried out ablation experiments on the PEMSD4 data set, and predicted the traffic conditions in the next 15mins, 30mins and 60mins respectively. The MAPE, MAE and RMSE indicators are used to measure the prediction accuracy, and the results are shown in Figure 6. MST-GNN has the highest improvement in the RMSE index, and as the prediction time increases, the accuracy advantage of MST-GNN becomes more obvious. This result validates the effectiveness of the multi-spatial-temporal-scale module in the MST-GNN model for the research question.

## 5 DISCUSSION AND CONCLUSION

In the paper, we propose MST-GNN model for traffic flow prediction. Aiming at the problem that existing models ignore the cross-temporal-scale and cross-spatial-scale characteristics of traffic data, our research proposes a simple but effective solution for scales fusion. Our model contains a multi-temporal-scale features fusion module as well as a multi-spatial-scale features fusion module. We conducted comparative experiments on two public datasets, and in both our model showed SOTA performance. In addition, we also tested the calculation time of the model to measure the operating efficiency of the model, and the experimental results prove that our model has high prediction efficiency. Although MST-GNN has unified two scales of space and time, our future direction will focus on integrating more scales knowledge.

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
