

Figure 1: Performance sensitivity analysis on time steps of coarse-grained data input. The metrics: (a) MAPE; (b) MAE; (c)RMSE.

## A PARAMETER SENSITIVITY ANALYSIS

To test the sensitivity of the input time step parameter for coarse time granularity data, experiments are conducted on the PMSED4 dataset. The result is shown in Figure 1. Overall, when the input step size increases, the prediction accuracy improves. And when the forecast period is longer, the improvement effect is more obvious. As shown in Figure 1, the improvement effect of the 60MINS prediction task is the most obvious. In order to test the effect of the model in the general situation, in the experiment, we uniformly input the coarse time granularity data of 12 time steps.

## B MODEL ROBUSTNES

The MST-GNN proposed in this paper uses data at two spatial scales, where the coarse-grained data is obtained by the Louvain community detection algorithm. Coarse-grained data obtained through other clustering methods may affect the predictive power of the model. In this subsection, we test the predictions of the model on PeMSD4 dataset when spectral clustering methods are used to obtain data at the region scale. The number of clusters used for spectral clustering was set to 31. The experimental results are shown in Table 1. Compared to the classical models presented in the paper, it has been observed that MST-GNN can achieve higher prediction accuracy even when utilizing spectral clustering methods for obtaining coarse-grained data. Therefore, MST-GNN can be used in conjunction with simple clustering methods and maintains high prediction accuracy with good robustness.

Table 1: Prediction results under spectral clustering on PeMSD4 dataset.

| Metric | MAPE | MAE | RMSE |
|---|---|---|---|
| 5mins | 0.10 | 14.87 | 23.86 |
| 15mins | 0.11 | 15.70 | 25.10 |
| 30mins | 0.124 | 17.60 | 27.76 |