# OpenReview forum: "MST-GNN: Graph Neural Network with Multi-Granularity in Space and Time for Traffic Prediction"
_ICLR.cc/2024/Conference — ICLR 2024 Conference Withdrawn Submission_

### Official Review · Reviewer_feuG · 2023-10-28

**Soundness:** 3 good
**Presentation:** 3 good
**Contribution:** 3 good
**Rating:** 5
**Confidence:** 4

**Summary:**

This paper proposes a graph learning model called MST-GNN, which captures the comprehensive behaviors and dynamics of traffic patterns by fusing multi-scale features from both space and time perspectives. The authors inject the multiscale spatio-temporal structure into a graph neural architecture with a tailored fusion module. Extensive experiments are conducted.

**Strengths:**

1. This paper is well-presented and well-organized.
2. This paper proposes a graph learning model called MST-GNN, which captures the comprehensive behaviors and dynamics of traffic patterns by fusing multi-scale features from both space and time perspectives.
3. Extensive experiments are conducted.

**Weaknesses:**

1.  This paper states that the SOTA ignore the traffic conditions at the urban regional scale, but there are many works like TrafficGAN, CurbGAN which predict traffic in fine-grained levels, can you compare the proposed model with more existing works?
2. For the second challenge---ignoring the integration of traffic features across time scales, the traffic data presents more dynamics in fine-grained series, and again, there are many works trying to capture the temporal dynamics of traffic data (e.g., cST-ML), can you further explain what the advantage is over the existing works.

**Questions:**

Please address the questions above.

---

### Official Review · Reviewer_M4FU · 2023-10-30

**Soundness:** 2 fair
**Presentation:** 2 fair
**Contribution:** 2 fair
**Rating:** 5
**Confidence:** 3

**Summary:**

The paper proposes a graph learning model called MST-GNN (Multi-Scale Traffic Graph Neural Network) for traffic flow prediction in Intelligent Transportation Systems (ITS). While deep learning approaches have shown superior performance in capturing spatio-temporal correlations in traffic conditions, most existing studies focus on predicting traffic at a single spatial scale, such as the road-segment level. However, according to the Hierarchy Theory, meaningful patterns can emerge at multiple levels of detail. Therefore, predicting traffic at an inappropriate scale can lead to a misunderstanding of features. To address this limitation, MST-GNN captures comprehensive behaviors and dynamics of traffic patterns by fusing multi-scale features from both space and time perspectives. It considers the larger-scale regional level, which is typically more relevant in ITS applications. Additionally, the model leverages the multi-scale structure inherent in time series data to fully capture dynamic traffic pattern features.

**Strengths:**

1. Multi-scale feature fusion: One of the strengths of MST-GNN is its ability to capture and fuse multi-scale features from both space and time perspectives. By considering traffic patterns at different spatial scales and incorporating the hierarchical nature of traffic processes, the model can capture more comprehensive behaviors and dynamics. This multi-scale feature fusion enables a more accurate representation of traffic conditions and improves the predictive performance of the model.

2. State-of-the-art accuracy: MST-GNN demonstrates state-of-the-art accuracy in traffic flow prediction on two benchmark datasets. By effectively incorporating the multi-scale spatio-temporal structure and leveraging the tailored fusion module within the graph neural architecture, the model achieves superior prediction results compared to existing approaches. This highlights the effectiveness of MST-GNN in capturing meaningful patterns and dynamics at different scales, leading to improved accuracy in traffic flow prediction tasks.

**Weaknesses:**

1. Computational complexity: One potential weakness of MST-GNN is its computational complexity. The fusion of multi-scale features from both space and time perspectives requires additional computations and memory resources. This increased complexity could result in longer training times and higher computational requirements, making the model less efficient, especially in real-time traffic prediction scenarios or when dealing with large-scale datasets. The scalability of the model may also be limited due to its computational demands.

2. Limited explanation of fusion module: The paper mentions a tailored fusion module within the graph neural architecture, but it does not provide detailed information on the specific design and implementation of this module. Without a clear explanation of how the fusion is performed and what mechanisms are employed, it becomes challenging to assess the effectiveness and interpretability of the fusion process. A more detailed description of the fusion module would enhance the clarity and reproducibility of the model, allowing for a better understanding of its inner workings and potential areas for improvement.

**Questions:**

Questions are provided in weaknesses.

**Details Of Ethics Concerns:**

None.

---

### Official Review · Reviewer_ZZx2 · 2023-10-30

**Soundness:** 3 good
**Presentation:** 2 fair
**Contribution:** 2 fair
**Rating:** 3
**Confidence:** 4

**Summary:**

This paper proposes a model named MST-GNN for traffic flow prediction. The core idea is to consider multi-scale traffic flow data in terms of both space and time, i.e., traffic flow data at single sensor location and single time interval (e.g., 5 minutes) vs. those at spatial regions and aggregate time intervals (e.g., 1 hour). Then, multiple GCNs are used to learn the spatial patterns at different spatial scales, and multiple LSTMs are used to learn the temporal patterns at different time scales. The learned patterns are fused with attention networks and fully connected layers to produce the final predictions. Experimental results on two PeMS datasets show the effectiveness of the proposed model.

**Strengths:**

1. The idea of multi-scale spatial and temporal pattern learning is interesting.

2. The proposed model is shown to be effective on real datasets.

**Weaknesses:**

1. While the idea of multi-scale spatial and temporal pattern learning is interesting, the model design is a straightforward combination of GCNs, LSTMs, and attention networks with limited technical contributions. There are no discussions on why it should be a two-level (single location vs. region, and 5 minutes vs. 1 hour) structure in both space and time, and there is no theoretical analysis on the effectiveness of the proposed idea.

2. The experimental results are not quite convincing. It is unclear why the paper chose to show prediction errors for 5, 15 and 30 minutes, while the literature typically shows results for 15, 30, 45, and 60 minutes.

In supplementary materials, there is an additional experiment on the impact of input time span. It is unclear why there are no results on varying the output time span.

Another experiment that would be interesting to see is to feed the raw fine-grained data into both branches of the MS-Blocks and both branches of the MT-Blocks, i.e., to verify if the performance gain was because of the use of data of multiple granularity or because of a more complex model.

3. Presentation issues:
"state-of-the-art accuracy prediction results" => "state-of-the-art prediction accuracy results"
"Traffic flow prediction is not a new task, early attempts at prediction were made..." => "Traffic flow prediction is not a new task. Early attempts at prediction were made..."
"is not same" => "is not the same"
"meaningful patterns may emergency" => "meaningful patterns may emerge"
missing whitespace: "community detectionClauset et al. (2004);" (there are many such cases in the paper)
"For node Gc in ni"

**Questions:**

See Weak Points 1 & 2.